# Enhanced Spontaneous Emission of CsPbI_3_ Perovskite Nanocrystals Using a Hyperbolic Metamaterial Modified by Dielectric Nanoantenna

**DOI:** 10.3390/nano13010011

**Published:** 2022-12-20

**Authors:** Hamid Pashaei Adl, Setatira Gorji, Andrés F. Gualdrón-Reyes, Iván Mora-Seró, Isaac Suárez, Juan P. Martínez-Pastor

**Affiliations:** 1Instituto de Ciencia de Materiales, Universidad de Valencia (ICMUV), 46071 Valencia, Spain; 2Institute of Advanced Materials (INAM), Universitat Jaume I, Avenida de Vicent Sos Baynat, s/n, 12071 Castello de la Plana, Spain; 3Facultad de Ciencias Instituto de Ciencias Químicas, Isla Teja, Universidad Austral de Chile, Valdivia 5090000, Chile; 4Escuela Técnica Superior de Ingeniería, Universidad de Valencia, 46100 Valencia, Spain

**Keywords:** CsPbI3, hyperbolic metamaterial, MIE resonator, Purcell effect

## Abstract

In this work, we demonstrate, theoretically and experimentally, a hybrid dielectric-plasmonic multifunctional structure able to provide full control of the emission properties of CsPbI_3_ perovskite nanocrystals (PNCs). The device consists of a hyperbolic metamaterial (HMM) composed of alternating thin metal (Ag) and dielectric (LiF) layers, covered by TiO_2_ spherical MIE nanoresonators (i.e., the nanoantenna). An optimum HMM leads to a certain Purcell effect, i.e., an increase in the exciton radiative rate, but the emission intensity is reduced due to the presence of metal in the HMM. The incorporation of TiO_2_ nanoresonators deposited on the top of the HMM is able to counteract such an undesirable intensity reduction by the coupling between the exciton and the MIE modes of the dielectric nanoantenna. More importantly, MIE nanoresonators result in a preferential light emission towards the normal direction to the HMM plane, increasing the collected signal by more than one order of magnitude together with a further increase in the Purcell factor. These results will be useful in quantum information applications involving single emitters based on PNCs together with a high exciton emission rate and intensity.

## 1. Introduction

In the last decade, metal halide perovskite nanocrystals (PNCs) have demonstrated excellent performances for the development of vanguardist optical sources. Synthetized under a cheap and straightforward colloidal chemistry, PNCs are tailor-made emitters whose bandgap is tunned with the composition through a broad UV-vis-NIR spectra (400–700 nm) [1,2] and whose photoluminescence quantum yield at room temperature can reach values up to 100% [3]. Moreover, their colloidal chemistry nature enables a straightforward incorporation on several substrates or optical architectures by simple dipping or coating methods [4,5]. However, lead-containing PNCs are toxic and typically characterized by a low stability under ambient conditions, which limits their future application in optoelectronics and photonics; these issues can be mostly solved by using encapsulation strategies in polymers or glasses [6,7]. In these conditions, PNCs have been successfully applied in several active devices, including lasers [8], light emitting diodes [9], and quantum sources [10]. Here, the appropriate design of photonic architecture where the PNCs are integrated can further enhance their optical properties and enable the manipulation of the emission rate. In this scenario, plasmonic metamaterials are disruptive photonic structures with unique electromagnetic states that allow light–matter interaction to be tailored at the nanoscale and hence provide full control of the light [11,12,13]. Particularly, hyperbolic metamaterials (HMMs) [14,15,16,17] have quickly gained a key role in nano-photonics due to their exceptional ability to enhance the spontaneous emission rate of a quantum emitter [18,19]. These extraordinary features result from the excitation of high-momentum electromagnetic states (high-k modes) overlapping the emitter [20,21]. However, since these modes become evanescent at the metamaterial’s surface, there is not a stablished solution to provide an efficient outcoupling of the emitted light. Under these conditions, the simultaneous enhancement of the optical signal together with a selective radiation towards a desired direction becomes a necessary task for this kind of device. At this stage, all-dielectric nanoantennas (ADNA), which enable both electric and magnetic MIE resonances, have emerged as nanostructures with excellent prospects for light manipulation [22]. Compared with their counterpart metallic nanoantennas, ADNA provide unidirectional radiation without the limitation of ohmic losses [23,24,25].

In this paper, we propose a hybrid system (HMM + MIE resonator + PNC) for increasing the spontaneous emission rate and PL intensity of CsPbI_3_ PNCs with emission at λ~720 nm. In particular, the structure consists of several alternating metal (Ag) and dielectric (LiF) layers covered by TiO_2_ dielectric spheres. This hybrid dielectric-plasmonic architecture presents high scattering efficiency and it is fabricated by simple deposition techniques. The resulting device demonstrates an efficient coupling of emitted light by the PNCs to both the scattering modes of the nanoantenna and the high-k modes of the HMM. First, the lifetime of excitons emitted by the CsPbI_3_ PNCs is reduced from τ∼1.75 ns (in the reference sample) to τ∼0.62 ns when they are coupled (d = 20 nm, with d being the separation between the PNC layer and the last metal layer of the HMM) to the optical modes of the HMMs. Furthermore, the incorporation of the MIE nanoresonators results not only in a redirection of the emitted light towards the normal direction to the HMM plane, but also in a further reduction in the lifetime, down to 0.45 ns, leading to an important improvement in the Purcell factor (from 2.8 to 3.9). To the best of our knowledge this is the first time where a hybrid structure composed of HMM and dielectric nanoantenna is used to control the emission properties of PNCs.

## 2. Experimental Methods

### 2.1. Synthesis of CsPbI_3_ Nanocrystals

CsPbI_3_ PNCs were synthesized following the hot-injection method [2,26]. All the reactants were used as received without an additional purification process. Briefly, a Cs-oleate solution was prepared by mixing 0.41 g of Cs_2_CO_3_ (Sigma-Aldrich, Lyon, France, 99.9 %), 1.25 mL of oleic acid (OA, Sigma-Aldrich, Lyon, France, 90 %), and 20 mL of 1- octadecene (1-ODE, Sigma-Aldrich, Lyon, France, 90%) which were loaded together into a 50 mL three-neck flask at 120 °C under a vacuum for 1 h under constant stirring. Then, the mixture was N_2_-purged and heated at 150 °C to reach the complete dissolution of Cs_2_CO_3_. The solution was stored under N_2_, keeping the temperature at 100 °C to prevent Cs-oleate oxidation. For the synthesis of CsPbI_3_ PNCs, 1.0 g of PbI_2_ (abcr, Karlsruhe, Germany, 99.999%) and 50 mL of 1-ODE were loaded into a 100 mL three-neck flask. The mixture was simultaneously degassed and heated at 120 °C for 1 h under constant stirring. Then, a mixture of 5 mL each of both pretreated (130 °C) OA and oleylamine (OLA, Sigma-Aldrich, Lyon, France, 98%) were separately added to the flask under N_2_, and the mixture was quickly heated at 170 °C. Simultaneously, 4 mL of Cs-oleate solution was quickly injected into the mixture, and then, the reaction was quenched to immerse the mixture into an ice bath for 5 s. In order to perform the isolation process of PNCs, the colloidal solutions were centrifuged at 4700 rpm for 10 min. Then, PNC pellets were separated after discarding the supernatant and redispersed in hexane to concentrate the PNCs at 50 mg/mL.

### 2.2. Fabrication and Characterization of the Device

The HMM structure is composed of alternating metal (Ag) and dielectric (LiF) films. The multilayer structure was fabricated using thermal evaporation under high vacuum (2 × 10^−5^ mbar) on top of a silicon wafer, which was preliminarily cleaned by following the procedure reported elsewhere [27]. TiO_2_ nanoresonators were deposited on the top of the HMM with the following procedure. First, the substrate was cleaned by spin coating (1000 rpm, 120 s), ethanol, isopropanol, and acetone, sequentially. Then, a spacer layer was formed by spin-coating a poly (methyl methacrylate) (PMMA) solution in toluene at 3000 rpm for 40 s and baking it at 120 °C for 2 min to avoid directly touching the emitter with the topmost metal layer of the HMM. Finally, the dielectric spheres were drop-casted on top of the structure after being dissolved in a mixture of water (20%) and ethanol (80%). It is worth noting that the water-to-ethanol ratio has been optimized for a uniform deposition. Here, the density of the deposited spheres on the surface was easily changed by changing the concentration of the spheres in the solution. Finally, after drying the samples, PNCs were deposited by dip coating (2 min) with a colloidal suspension of PNCs 1 mg/mL in hexane.

The specular reflectance of the same HMM structures was measured at single wavelengths using different lasers by a homemade setup incorporating a goniometer for the HMM holder and detector (θ–2θ configuration). For low-temperature photoluminescence (PL) and timeresolved PL (TRPL) measurements, the samples were held in the cold finger of a closed-cycle He cryostat (ARS DE-202), which can be cooled down to approximately 15 K. PL was excited with a 200 fs pulsed Ti: sapphire passive mode-locked laser (Coherent Mira 900D, 76 MHz repetition rate, San José, United States) operating at a wavelength of 810 nm and doubled to 405 nm using a BBO crystal. The PL signal was dispersed using a double 0.3 m focal length grating spectrograph (Acton SP-300i from Princeton Instruments, United States) and detected with a cooled Si CCD camera (Newton EMCCD from ANDOR) for PL measurement and with a silicon single-photon avalanche photodiode (micro photon device) connected to a time-correlated single-photon counting electronic board (TCC900 from Edinburgh Instruments, United States) for TRPL measurements. These measurements were carried out the same day after preparation of the samples at 15 K in vacuum atmosphere (the cryostat), hence any degradation of PNCs was observed in these conditions, assuring the reproducibility of our results. Moreover, the laser excitation fluence was limited with neutral density filters down to 165 nJ/cm^2^ to assure the stability of the PNCs during these experiments.

The XRD characterization was performed with a D4 Endeavor diffractometer from Bruker-AXS, using a CuKα radiation source (λ = 1.54056 Å) with the following conditions: 2θ range of 5–70° (0.02°/step and 1.2 s/step).

### 2.3. Device Design and Simulations

The far field and near field emitted by the device were simulated with the RF module of 2D COMSOL Multiphysics. A perfectly matched layer (PML) enclosing the photonic structure was established to block unwanted reflections from the computational domain’s outside edges. The far field emitted by the device was calculated using the Lorentz reciprocity theorem which declares that the relationship between a localized oscillating current and the consequential electric field is unaffected if the positions where the current is placed and where the field is measured are swapped [28,29].
(1)$$\int \int \int \;{\vec{E}}_{J1}.\;{\vec{J}}_{2}d^{3}\;\vec{r}=\int \int \int \;{\vec{E}}_{J2}.\;{\vec{J}}_{1}d^{3}\;\vec{r}$$
where $\vec{J}$ is the localized time-harmonic current density oscillating at an angular frequency of ω and $\vec{E}$ is the consequential electric field resulting from this current density.

In the simulations, the permittivity of lithium fluoride (LiF) was taken to be ε_LiF_ = 1.95, whereas those of silver (Ag), silicon (Si), and TiO_2_ spheres were taken from Refs. [30,31,32].

## 3. Results and Discussion

### 3.1. Design and Simulations

#### 3.1.1. Description of the Device

It can be clearly observed that our CsPbI_3_ PNCs showed only the (100) and (200) diffraction peaks of the standard cubic-phase perovskite structure (crystallographic data ICSD-161481), as observed in the XRD spectrum of Figure 1a. The HMM used in this work (Figure 1b) was designed to allow hyperbolic permittivity dispersion at λ ≥ 370 nm and, with it, to improve the radiative rate of the CsPbI_3_ PNCs deposited atop (Figure 1c) [16]. It consists of six periods of alternating metal (Ag) and dielectric (LiF) layers with thicknesses of 25 and 35 nm, respectively. The device is capped by a PMMA layer that acts as spacer between the HMM and PNCs. In these conditions, the HMM enhances the spontaneous emission rate of CsPbI_3_ PNCs by the weak coupling of the exciton with the modes allowed in the multilayer structure [16]. However, most of the light generated at the PNCs-HMM cannot be efficiently outcoupled, because most of the emission is contributing to surface waves travelling longitudinally through the structure. In this way, the device is complemented with TiO_2_ dielectric spheres of 270 nm deposited on the HMM (see optical microscope image of Figure 1d). The TiO_2_ nanoresonators of such a size present a fundamental MIE resonance at around 700 nm [33], which is close to the emission wavelength of our PNCs. In these conditions, the TiO_2_ nanoantennas are able to scatter the emitted light from PNCs towards the normal direction to the HMM plane and with it, to further improve the exciton radiative rate [32].

#### 3.1.2. Design of Spacer

The thickness d of the spacer, defined as the distance between the metal-terminated HMM surface and the PNC layer, is a critical parameter that provides the optical coupling of the emitters with the HMM. Considering the results published elsewhere [16], two samples were fabricated with nominal spacer thicknesses d = 20 nm and d = 250 nm. The thicker one (d = 250 nm) is expected to provide a negligible coupling between the exciton and the HMM and is used here as a reference. On the contrary, the spacer d = 20 nm was chosen to give an appreciable overlap of the HMM modes with the active material, hence providing a remarkable enhancement of the emission rate (~3 in [16]).

The emission of PNCs was modelled here as classical forced electric dipole oscillators distributed in the active layer [28,34,35], where each isolated dipole presented its own orientation and frequency of emission. To accurately simulate the influence of the HMM on the emission of multiple dipoles composing the active layer one needs to consider the multiple reflection between the different layers and the incorporation a of large number of dipoles. Since this simulation required extensive computational resources, we exploited the Lorentz reciprocity theorem [29,36] to quantitatively characterize the spatial emission pattern of the PNCs–HMM samples (see methods section). The main idea of this approximation (the reciprocity theorem) is to convert a light out-coupling problem of a structure into a light in-coupling problem [37,38,39]. This approximation significantly simplifies the simulation without losing accuracy and provides the needed information of the spatial emission pattern in computationally efficient calculations [29,36].

Following these approximations, we were able to calculate the far-field intensity of vertically and horizontally oriented dipoles (Figure 2a,b, respectively) deposited on the top of the HMM. Red and blue colors refer to a spacer thickness of 20 nm and 250 nm, respectively. The shadow area of the images indicates the solid angle launched in a standard collection system with a numerical aperture of 0.26 (angles between −15° and 15°). On one hand, the emission for a vertically oriented dipole (Figure 2a) was out of the numerical aperture of our collection system (cone of ± 15°), hence these dipoles did not contribute to the collected signal. On the other hand, a horizontally oriented dipole (Figure 2b) presented a big overlap with the collection area. In particular, ≈ 100 % of the light emitted by the dipole deposited on the thicker spacer, d = 250 nm, was practically launched inside the collection area (light dispersed in a cone of ± 30°), while the overlap was importantly reduced (cone of ± 70°) with the thinner spacer, d = 20 nm. Under these conditions, there was a compromise between the Purcell factor, obtained with d = 20 nm, and the redirection of the emitted light, improved with a spacer d = 20 nm.

#### 3.1.3. Effect of the MIE Resonator

From the previous simulation we inferred that the device could be improved by redirecting the emitted light towards the normal direction to the HMM surface. For this purpose, we proposed the use of TiO_2_ nanospheres as excellent scatterers (i.e., nanoantenna) at the wavelengths of interest (around the PL emission of the PNCs).

Figure 3 simulates (using COMSOL, see methods) the influence of the TiO_2_ nanoantenna on the electric field distribution of a horizontal dipole deposited on the top of the HMM structure. Figure 3a,b plots the enhancement factor of the near field (Γ) as a function of the wavelength for spacers d = 250 nm and d = 20 nm. The emission enhancement factor (Γ) was calculated by dividing the electric field distribution of the hybrid structures with 250 and 20 nm spacers with the electric field of the reference sample (HMM structure with 250 nm spacer and without MIE nanoresonators). For d = 20 nm, Γ = 9 at the spectral range of emission of the PNCs (λ = 700–740 nm), while Γ < 3 for the case of d = 250 nm in the whole spectra. Figure 3c–f shows the effect of the nanosphere on the spatial distribution of the near-field emission. The green arrows show the direction of the Poynting vector on a logarithmic scale. Clearly, the TiO_2_ nanospheres deposited onto the thinner spacer layer concentrated the emitted light towards the normal direction to the HMM plane by redirecting the dipole light emission. For the spacer d = 250 nm, the incorporation of the MIE nanoresonators only produced a small improvement in the redirection of the emitted light (Figure 3c,d). On the other hand, for d = 20 nm the nanoantenna produces a stronger concentration of the emitted light along the normal direction (Figure 3e) compared to the case where no MIE nanoresonators were present (Figure 3f).

### 3.2. Optical Characterization and the Experimental Setup

Figure 4a illustrates the experimental setup for PL and TRPL measurements in backscattering geometry. In this experiment, samples were held in the cold finger of a closed-cycle He cryostat to tune the temperature in the range 15–300 K. A long working distance 10× microscope objective with a numerical aperture of NA = 0.26 was mounted outside the cryostat to collect and send the emitted light to the spectrometer/spectrograph system (more information can be found in the experimental methods section). We developed the experiments at 15 K to prevent the influence of the carrier trapping/detrapping effect on the exciton recombination dynamics by increasing temperature [16]. At room temperature, the PL decay kinetics in PNCs was masked by shallow non-quenching traps, which resulted in longer recombination times [40,41], while at cryogenic temperatures the participation of shallow traps in the formation of the PL decay kinetics was prevented, and the PL decay was mainly due to the exciton radiative recombination.

The PL peak wavelength in the reference sample (d = 250 nm) was centered at 717 nm (Figure 4b), in agreement with the PL reported for these emitters [42]. Nevertheless, when the PNCs were deposited on the HMM structure with a spacer of d = 20 nm, the PL experienced a redshift of 3 nm that can be attributed to the interaction of the exciton with the HMM modes, as previously reported [16]. This PL redshift was accompanied by a slight decrease in the PL intensity due to the preferential emission of light into the high-k metamaterial modes, which travels parallel to the HMM surface. More importantly, we observed a shortening of the measured PL decay (τ) when reducing the PMMA spacer thickness from 250 to 20 nm (Figure 4c). The TRPL decays were fitted with a biexponential decay function [43]. In particular, the exciton lifetime in the sample with d = 250 nm was τ ∼ 1.75 ns, which is very close to values reported by other authors [41,44]. As expected, the exciton lifetime measured in the sample with d = 20 nm now was as low as τ ∼ 0.62 ns, which is ascribed to a Purcell factor of around 2.8 introduced by the HMM structure.

Figure 5 shows the PL spectra and transients of PNCs when they were deposited on the HMMs + nanoantenna system for spacers d = 250 nm and 20 nm. For the reference sample (d = 250 nm), the PL peak was again centered at λ = 717 nm (Figure 5a), in agreement with the PL measured in the reference sample without TiO_2_ nanoantenna (Figure 4b). However, the collected PL signal was slightly increased and experienced a lifetime shortening from τ ∼ 1.75 to 1.35 ns (Figure 5b). This was due to the Purcell effect introduced by the coupling of the PNC exciton to the nanoantenna, that is, introducing a Purcell factor ∼ 1.3, affecting both the PL intensity and decay.

The results, however, were more impressive for d = 20 nm, where the emission of the PNCs was enhanced dramatically by the combination of both coupling systems (HMM and MIE nanoresonators). First, the collected PL intensity (Figure 5c) was more than one-fold that measured for the similar sample (d = 20 nm) without the MIE resonator (Figure 4b). Moreover, the lifetime was further reduced to τ ∼ 0.45 ns (Figure 5d), which represents a further Purcell enhancement of around 1.4 with respect to the case without the TiO_2_ nanospheres in Figure 4c, where τ ∼ 0.62 ns, as referred to above for the reference samples with d = 250 nm. Clearly, the incorporation of the TiO_2_ spherical nanoantenna improved the light collected by two different factors: (i) a further increase in the Purcell factor derived by the coupling of the emitter with the scattering modes, and (ii) a selective reorientation of the emitted light towards the normal direction to the HMM plane [32,45,46]. Factor (i) gave rise to an overall/total Purcell factor of around 3.9 (2.8 without nanoantenna) and factor (ii) lead to compensation of the PL intensity decrease (due to the preferential emission of light into the high-k HMM modes) and a further intensity increase due to the important angle squeezing of emitted light towards the normal direction to the HMM surface.

## 4. Conclusions

To summarize, dielectric (TiO_2_) spherical nanoresonators were integrated with HMM substrates in order to enhance the exciton radiative rate and emission intensity of CsPbI_3_ PNCs. The coupling of excitons photogenerated in CsPbI_3_ PNCs to the optical modes of these HMM substrates reduced the lifetime duration from τ∼ 1.75 ns (reference) to τ∼ 0.62 ns (d = 20 nm). Furthermore, the incorporation of the TiO_2_ nanoantenna results in both the redirection of the emitted light towards the normal direction to the HMM surface together with the induction of an additional Purcell effect, because the lifetime is further shortened to τ ∼ 0.45 ns. More important is the compensation of the PL intensity decrease observed in the HMM–PNC system, which is due to the preferential emission of light into the high-k HMM modes, and the further increase in the PL intensity due to the redirection of emitted light towards the direction perpendicular to the HMM surface. These results will have important implications for future research into single-photon sources and other quantum photonic applications of PNCs because of the increased exciton radiative rate and emission intensity.

## Figures and Tables

**Figure 1 nanomaterials-13-00011-f001:**
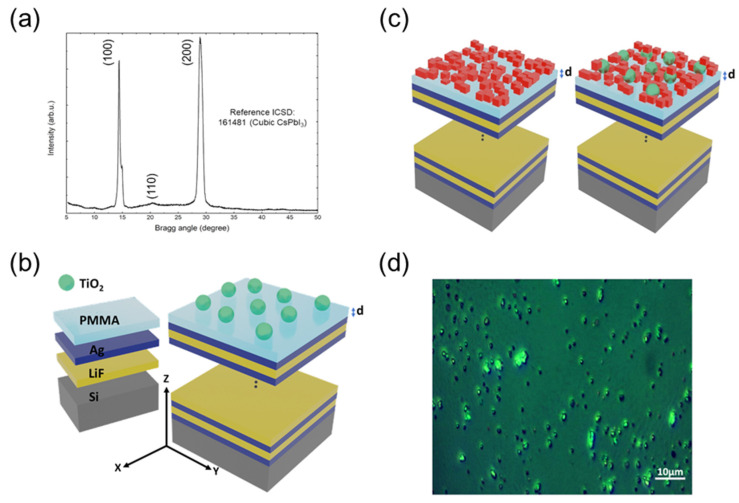
(**a**) XRD spectrum and analysis of CsPbI_3_ PNCs. (**b**) Schematic representation of the HMM device with TiO_2_ nanoresonators (green spheres) atop. (**c**) Schematic representation of HMM + PNCs (red cubes) and HMM + PNCs + nanoantenna, with a spacer layer of thickness d between them. (**d**) Optical microscope image of TiO_2_ nanospheres dispersed on top of the HMM structure.

**Figure 2 nanomaterials-13-00011-f002:**
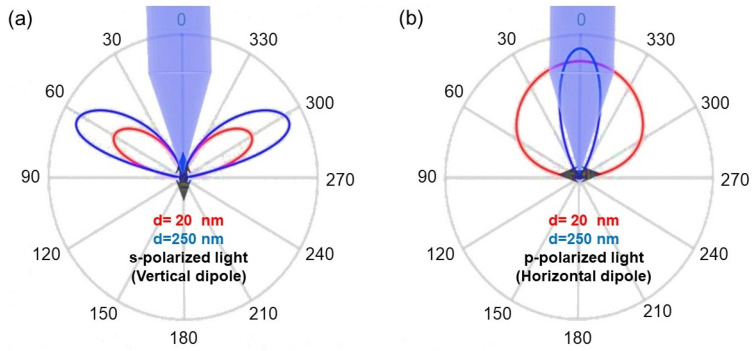
The angular distribution of the radiated intensity in the far-field for: (**a**) a vertically and (**b**) a horizontally oriented point dipole. The blue and red curves correspond to the thin (d = 20 nm) and thick (d = 250 nm) spacer layers, respectively, and the black double arrows show the point dipole’s orientation. The numerical aperture of the objective used for light collection is represented schematically by the blue shaded cones.

**Figure 3 nanomaterials-13-00011-f003:**
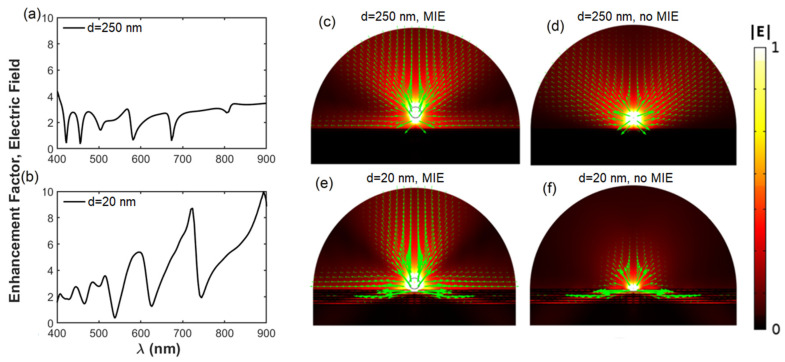
Effect of the TiO_2_ nanoantenna on the enhancement factor and field distribution. Calculation of the enhancement factor spectra with a spacer of (**a**) d = 250 and (**b**) d = 20 nm. Distribution of the normalized electric field |E| of a horizontal (p-polarized) dipole placed at the top of the d = 250 nm spacer layer with (**c**) and without (**d**) the MIE nanoresonator. (**e**,**f**) corresponds to (**c**,**d**) but with a thinner spacer layer (d = 20 nm). The point source is considered to emit at λ = 720 nm and the green arrows show the direction and intensity of the energy flux on a logarithmic scale.

**Figure 4 nanomaterials-13-00011-f004:**
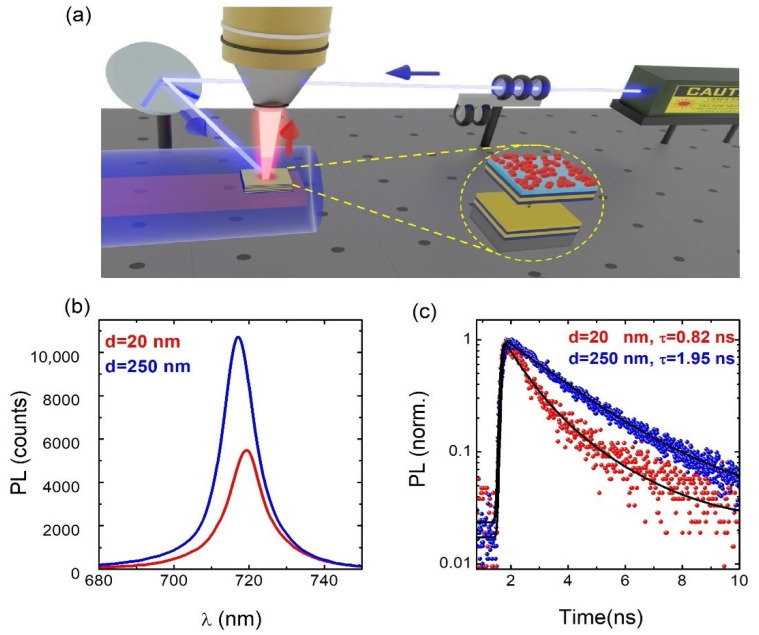
(**a**) Illustration of the experimental setup for measuring the PL and TRPL spectra of PNCs at low temperatures. (**b**) PL spectra of CsPbI_3_ PNCs deposited on top of HMM substrates (PNC+HMM) with two different spacer thicknesses, d = 20 nm (red curve) and d = 250 nm (blue curve). (**c**) PL transients of CsPbI_3_ PNCs measured at PL peak wavelengths of spectra in (**b**) and two-exponential fittings (black continuous curves). PL and PL transients are measured at 15 K.

**Figure 5 nanomaterials-13-00011-f005:**
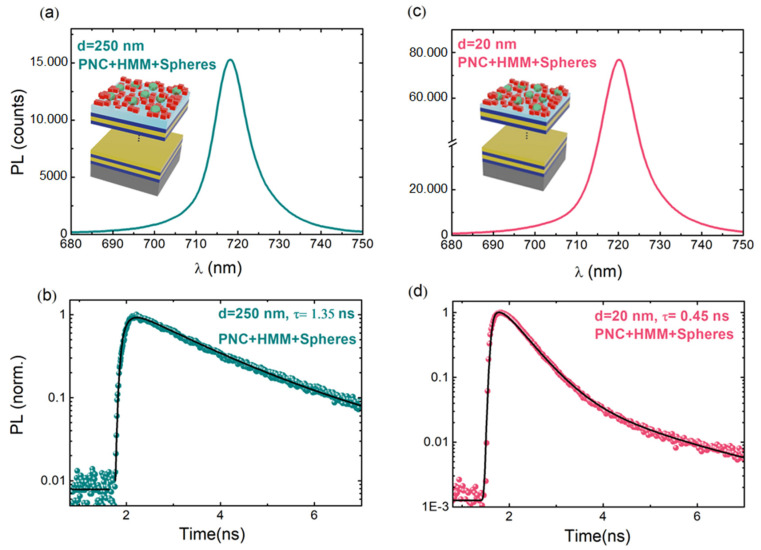
(**a**,**c**) The effect of TiO_2_ nanospheres (270 nm in diameter) incorporated into HMM structures with d = 250, 20 nm spacer layers on the PL emission of CsPbI_3_ PNCs. (**b**,**d**) PL transients of the same PNCs corresponding to PL peak wavelengths of spectra in (**a**,**c**), respectively. All measurements were performed at 15 K.

## Data Availability

Not applicable.

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
