# Peer review of "Enhanced Spontaneous Emission of CsPbI3 Perovskite Nanocrystals Using a Hyperbolic Metamaterial Modified by Dielectric Nanoantenna"

_nanomaterials, 2022, doi:10.3390/nano13010011_

Round 1

Reviewer 1 Report

Reviewer comments:

In this work, the authors report on the enhanced PL of CsPbI3 PNCs upon developing a multifunctional architecture consisting of a hyperbolic metamaterial and TiO2 sphere resonators. The introduction of the latter is found to increase the PL intensity upon the redirection of emitted light perpendicularly to the surface. The paper is well written, while presenting interesting results for the reader. However, a few points need to be clarified and addressed. Thus, in order to be considered for publication in Nanomaterials (MDPI), the authors are strongly encouraged to address the points listed below (major revisions) that are critical for strengthening the paper.

 Suggested revisions and recommendations:

1. Page 1, lines 1-9: The authors refer to the advantages of PNCs for a wide range of photonic and optoelectronic applications. However, it is important to refer also to the two main disadvantages of perovskite-based materials, namely the low stability, and the high lead toxicity. Following this, the authors are kindly asked to refer to the two provided solutions for resolving these problems, i.e. encapsulation within polymers and glasses (S. N. Raja et al., ACS Appl. Mater. Interfaces 8, 35523, 2016, and I. Konidakis et al., Nanoscale 14, 2966, 2022). These approaches render the application of PNCs plausible for realistic employment in photonic and optoelectronic applications.

 2. Page 2, Section 2.1, first line: References [24] and [25] need to be placed before the full stop.

 3. Page 2, Section 2.2: The authors are kindly asked to include if possible, the XRD patterns of the so-formed PNCs.

 4. Page 4, Figures 1a and b: The text and symbols within these figures are not readable. Also, are the red dots in b PNCs? It will be good to add a caption for this. Is there the corresponding optical microscopy photo available after the addition of PNCs? If yes, it will be good to show this photo as well.

 5. Page 7, Figure 4: The authors need to add in the caption that the PL spectra of b are measured at 15 K. Are the PL dynamics shown in c also measured at 15 K? Please clarify this in the caption.

 6. Page 7, Section 3.2: Did the author perform any PL and PL decay kinetics stability tests for the developed devices upon exposure to laboratory air and moisture? How do these degrade with time over exposure?

 7. Page 7, last three lines: The authors mention that the exhibited lifetime of the PNCs is of 1.75 ns and that it is comparable to the one of freshly prepared samples. Does this change upon exposure to humidity? Please refer to comment above.

 8. Page 8, Figure 5: The authors need to correct ‘sphere’ or ‘spheres’ within the four graphs, so that there is a uniform notation throughout the figure.

Author Response

Reviewer 1:

In this work, the authors report on the enhanced PL of CsPbI3 PNCs upon developing a multifunctional architecture consisting of a hyperbolic metamaterial and TiO2 sphere resonators. The introduction of the latter is found to increase the PL intensity upon the redirection of emitted light perpendicularly to the surface. The paper is well written, while presenting interesting results for the reader. However, a few points need to be clarified and addressed. Thus, in order to be considered for publication in Nanomaterials (MDPI), the authors are strongly encouraged to address the points listed below (major revisions) that are critical for strengthening the paper.

We acknowledge the reviewer for his/her positive and constructive comments. In the following lines we address all these interesting criticisms.

We have revised the manuscript and introduced some minor corrections (some words) throughout the text (possibly some of them are not highlighted in yellow), mainly related to the correct use of nanoresonators, nanospheres, nanoantenna. We have also shortened a bit the title, now Enhanced spontaneous emission of CsPbI3 perovskite nanocrystals using a hyperbolic metamaterial modified by dielectric nanoantenna.

Suggested revisions and recommendations:

  1. Page 1, lines 1-9: The authors refer to the advantages of PNCs for a wide range of photonic and optoelectronic applications. However, it is important to refer also to the two main disadvantages of perovskite-based materials, namely the low stability, and the high lead toxicity. Following this, the authors are kindly asked to refer to the two provided solutions for resolving these problems, i.e. encapsulation within polymers and glasses (S. N. Raja et al., ACS Appl. Mater. Interfaces 8, 35523, 2016, and I. Konidakis et al., Nanoscale 14, 2966, 2022). These approaches render the application of PNCs plausible for realistic employment in photonic and optoelectronic applications.

Answer: We thank the reviewer for this observation. Now the text has been modified to address these two major drawbacks of perovskite materials, as well as the solutions demonstrated in the reviewer suggested references. In particular we have added the following sentence in the introduction:

However, lead-containing PNCs are toxic and typically characterized by a low stability under ambient conditions, which limits their future application in optoelectronics and photonics; these issues can be mostly solved by using encapsulation strategies in polymers or glasses [6, 7].

  1. Page 2, Section 2.1, first line: References [24] and [25] need to be placed before the full stop.

Answer: We thank the reviewer for this observation. Now the references are placed before the full stop.

  1. Page 2, Section 2.2: The authors are kindly asked to include if possible, the XRD patterns of the so-formed PNCs.

Answer: We acknowledge the reviewer for this comment because the XRD provides interesting information about the nanocrystals under study. We have now included the pattern in Figure 1a, the description of the experiment in the experimental method and the conclusions in the information obtained from these patterns at the beginning of section 3.1.1.  with the following sentences:

(in section 2.2) The XRD characterization was performed through a D4 Endeavor diffractometer from Bruker-AXS, using a CuKα radiation source (λ = 1.54056 Å) with the following conditions: 2θ range of 5°-70° (0.02 °/step and 1.2 s/step). 

(in section 3.1.1) It can be clearly observed that these α-CsPbI3 NCs show only the (100) and (200) diffraction peaks of the standard cubic-phase perovskite structure (crystallographic data ICSD-161481).

  1. Page 4, Figures 1a and b: The text and symbols within these figures are not readable. Also, are the red dots in b PNCs? It will be good to add a caption for this. Is there the corresponding optical microscopy photo available after the addition of PNCs? If yes, it will be good to show this photo as well.

Answer: Figures 1a and 1b (now 1(b) and 1(c) have been modified by increasing the text font size and adding a more complete figure caption:

Figure 1. (a) XRD spectrum and analysis of CsPbI3 PNCs (b) Schematic representation of the HMM device with TiO2 nanoresonators (green spheres) atop. (c) Schematic representation of HMM + PNCs (red cubes) and HMM + PNCs + nanoantenna, with a spacer layer of thickness d between them. (d) Optical microscope image of TiO2 nanospheres dispersed on top of the HMM structure.

  1. Page 7, Figure 4: The authors need to add in the caption that the PL spectra of b are measured at 15 K. Are the PL dynamics shown in c also measured at 15 K? Please clarify this in the caption.

Answer: The caption has been adjusted …. (c) PL transients of CsPbI3 PNCs measured at PL peak wavelengths of spectra in (b) and two-exponential fittings (black continuous curves). PL and PL transients are measured at 15 K.

  1. Page 7, Section 3.2: Did the author perform any PL and PL decay kinetics stability tests for the developed devices upon exposure to laboratory air and moisture? How do these degrade with time over exposure?

Answer: We thank the reviewer again because this is a pertinent and important question, because of the stability issue in perovskite nanocrystals. After many preliminary experiments (done for the present papers and others, for example those in Refs 5, 16 and 17 – new numeration) we observed that PL spectra and decay times suffer of some evolution (see Ref. 5 for deeper explanation) after the first day of the sample preparation (deposition of PNCs atop the HMM or other substrates). For this reason, we did all experiments the first day where the sample was prepared (this was indicated in our previous paper, Ref. 16). Moreover, we are always using in our PL experiments at 15 K very low excitation densities, around 165 nJ/cm2. This low excitation power, together with the use of vacuum atmosphere and the low temperature, is preventing any degradation and the PL stability was assured during the experiments. We have now included the following sentence in the experimental methods:

These measurements were carried out the same day after preparation of the samples at 15 K in vacuum atmosphere (the cryostat), hence any degradation of PNCs is observed in these conditions and assuring the reproducibility of our results. Moreover, the laser excitation fluence was limited with neutral density filters down to 165 nJ/cm2 to assure the stability of PNCs during these experiments.

  1. Page 7, last three lines: The authors mention that the exhibited lifetime of the PNCs is of 1.75 ns and that it is comparable to the one of freshly prepared samples. Does this change upon exposure to humidity? Please refer to comment above.

Answer: See previous explanations for answering Question 6, where we addressed these issues: effect of ambient conditions on the PL of PNCs and stability of PL due to possible photodegradation (eventually under ambient conditions). As aforementioned both issues are avoided because of the use of low excitation power and experiments developed at 15 K in a cryostat (vacuum atmosphere). Moreover, the development of experiments always the same day where samples were prepared, introduced further reproducibility. The sentences proposed above should be now clarifying the concerns raised by the referee and hence it is not necessary to use the term “freshly” in that sentence.

  1. Page 8, Figure 5: The authors need to correct ‘sphere’ or ‘spheres’ within the four graphs, so that there is a uniform notation throughout the figure.

Answer: The plot label of the Figure 5 has been corrected accordingly.

Reviewer 2 Report

In the manuscript “Enhanced spontaneous emission of CsPbI3 perovskite nanocrystals using hybrid metal-dielectric sub-micrometric antennas with modified hyperbolic metamaterials” the authors investigate the perovskite nanocrystal photoluminescence enhancement by the combined effect of a dielectric layered metamaterial structure and MIE resonators. The work closely follow a previously published results by the same authors (ACS Photonics 2020, 7, 11, 3152–3160), but improves the outcomes by the addition of MIE scattering centers at the top metamaterial surface, providing a striking enhancement in the photoluminescence directionality. Moreover, the Purcell factor is further enhanced compared to the metamaterial beneficial effect alone, of about 35%.

I think the topic is of interested for Nanomaterials, the results are compelling and the data support the conclusions. The scholar presentation is good. I recommend the article to be published after minor corrections are noted.

I have two main comments for the authors:

1.      Why the diameter of the TiO2 sphere was chosen to be 270nm? Can you please comment on the expectation of your results considering a different nanoparticles size?

2.      Page 8: “First, the collected PL intensity, see Figure 5(c), is more than one-fold than that measured for the similar sample (d = 20 nm) without the MIE resonator (Figure 4(b)).” Assuming the laser power is constant throughout the experiments (if not, must be specified and it would be good to have the excitation power density value anyway), I see a 10-fold increase: 6000 to 80000 counts. Can the authors check this?

Here I note some minor corrections:

3.      Abstract: “The incorporation of TiO2 resonators deposited on the top of the HMM is able to counteract such undesirable through the coupling between the exciton and the MIE modes of the dielectric resonators.” I think a word is missing after “undesirable”

4.      Figure 5: “Spheres” in the plot label appears to be with and without “s”.

Author Response

In the manuscript “Enhanced spontaneous emission of CsPbI3 perovskite nanocrystals using hybrid metal-dielectric sub-micrometric antennas with modified hyperbolic metamaterials” the authors investigate the perovskite nanocrystal photoluminescence enhancement by the combined effect of a dielectric layered metamaterial structure and MIE resonators. The work closely follow a previously published results by the same authors (ACS Photonics 2020, 7, 11, 3152–3160), but improves the outcomes by the addition of MIE scattering centers at the top metamaterial surface, providing a striking enhancement in the photoluminescence directionality. Moreover, the Purcell factor is further enhanced compared to the metamaterial beneficial effect alone, of about 35%.

I think the topic is of interested for Nanomaterials, the results are compelling and the data support the conclusions. The scholar presentation is good. I recommend the article to be published after minor corrections are noted.

We acknowledge the reviewer for his/her positive and constructive comments. See below our answers.

We have revised the manuscript and introduced some minor corrections (some words) throughout the text (possibly some of them are not highlighted in yellow), mainly related to the correct use of nanoresonators, nanospheres, nanoantenna. We have also shortened a bit the title, now Enhanced spontaneous emission of CsPbI3 perovskite nanocrystals using a hyperbolic metamaterial modified by dielectric nanoantenna.

I have two main comments for the authors:

  1. Why the diameter of the TiO2 sphere was chosen to be 270nm? Can you please comment on the expectation of your results considering a different nanoparticles size?

Answer: We thank the reviewer for this important comment. The MIE resonator used in this work where commercial TiO2 nanospheres with a diameter of 270 nm. The TiO2 nanospheres of such a size presents the fundamental MIE resonance at around 700 nm (see, for example, Ad. Functional Materials 2018, 28, 1801958), which is close to the PL peak of the PNCs. Since the wavelength of the corresponding fundamental resonance red-shifts with the size of the sphere, we expect that the efficiency of the device will be worsen for much bigger and smaller sizes. We have now included the following sentence in the section 3.1.1. to clarify this issue: The TiO2 nanoresonators of such a size presents a fundamental MIE resonance at around 700 nm [34], which is close to the emission wavelength of our PNCs.  [REF 34: Ad. Functional Materials 2018, 28, 1801958].

2.Page 8: “First, the collected PL intensity, see Figure 5(c), is more than one-fold than that measured for the similar sample (d = 20 nm) without the MIE resonator (Figure 4(b)).” Assuming the laser power is constant throughout the experiments (if not, must be specified and it would be good to have the excitation power density value anyway), I see a 10-fold increase: 6000 to 80000 counts. Can the authors check this?

Answer: We thank the reviewer for this comment. The laser power remained constant in all the experiments (165 nJ/cm2), and the important enhancement of the PL intensity (from 6000 to 80000 counts) is originated by the incorporation of TiO2 spherical resonators, which resulted in both the redirection of the emitted light towards the normal direction to the HMM surface and the induction of an additional Purcell effect. This is one of the main conclusions of the manuscript.

Here I note some minor corrections:

3.Abstract: “The incorporation of TiO2 resonators deposited on the top of the HMM is able to counteract such undesirable through the coupling between the exciton and the MIE modes of the dielectric resonators.” I think a word is missing after “undesirable”.

Answer: The missing word has been added.

4.Figure 5: “Spheres” in the plot label appears to be with and without “s”.

Answer: The plot label of the Figure 5 has been corrected.

Round 2

Reviewer 1 Report

The authors have addressed all the points.